# 'Can you please hold my hand too, not only my breast?' The experiences of Muslim women from Turkish and Moroccan descent giving birth in maternity wards in Belgium

Liesbet Degrie[1], Bernadette Dierckx de Casterlé[2], Chris Gastmans[1], Yvonne Denier[1] *

1 Department of Public Health and Primary Care, Centre for Biomedical Ethics and Law, University of Leuven, Leuven, Belgium, 2 Department of Public Health and Primary Care, Academic Centre for Nursing and Midwifery, University of Leuven, Leuven, Belgium

* Yvonne.Denier@kuleuven.be

**Data Availability Statement:** Data cannot be made publicly available for ethical and legal reasons, since public availability would compromise patient

## Abstract

### Objectives

To reach nuanced understanding of the perinatal experiences of ethnic minority women from Turkish and Moroccan descent giving birth in maternity wards in Belgium thereby gaining insight into the underlying challenges of providing intercultural care for ethnic minority persons in a hospital setting.

### Methods

A qualitative study design was used by conducting In-depth interviews with 24 women from Turkish and Moroccan descent who gave birth during the past three years in maternity wards in Flanders, Belgium. The interviews were analysed using a Grounded Theory Approach.

### Results

This study shows that the women's care experiences were shaped by the care interactions with their caregivers, more specifically on the attention that was given by the caregivers towards two essential dimensions of the care relationship, viz. *Ereignis* (attention to *what* happens) and *Erlebnis* (attention to *how* it happens). These two dimensions were interrelated in four different ways, which defined the women's care experiences as being either '*uncaring*', '*protocolized*', '*embraced*' or '*ambiguous*'. Moreover, these experiences were fundamentally embedded within the women's *cultural* context, which has to be understood as a relational process in which an emotional and moral meaning was given to the women's care expectations, interactions and interpretations of care.

### Conclusions

The findings reveal that the quality of intercultural care depends on the nature and quality of care interactions between ethnic minority patients and caregivers much more than on the

confidentiality and participant privacy, and would violate the informed consent agreements made with the participants. Data are available upon request to the Social and Societal Ethics Committee (SMEC) of KU Leuven (contact via smec@kuleuven.be), which granted approval of the research project (G 2016 03 531).

**Funding:** This work was supported by the Fund for Scientific Research Flanders [grant number: G0B0914N]. The funding source had no involvement in decision during the study design, collection, analysis, writing or submission for publication.

**Competing interests:** The authors have declared that no competing interests exist.

way in which cultural questions and tensions are being handled or dealt with in a practical way. As such, the importance of establishing a meaningful care relationship should be the priority when providing intercultural care. In this, a shift in perspective on 'culture' from being an 'individual culture-in-isolation' towards an understanding of culture as being inter-relational and emerging from within these care relationships is necessary.

## Introduction

Despite the increasing multi-ethnicity within societies worldwide, ethnic minority patients still face significantly higher risk of being confronted with lower quality of healthcare, lower health outcomes, inequalities, disparities and barriers in access to care. [1–3] Healthcare services are being challenged to provide care for ethnic minority patients because of the increasing heterogeneity in their health determinants, needs and vulnerabilities. [3] Notwithstanding these challenges, the World Health Organization (WHO) recommends that healthcare services should ensure culturally appropriate care for every (ethnic minority) patient. [3]

Providing culturally appropriate care is particularly challenging in the hospital setting because of its acute, necessary and inevitable character. [4] Previous research has shown that lower health literacy and socioeconomic status in ethnic minority groups, budgetary restrictions in the hospital, difficulties in communication and differences in cultural interpretations of illness, health and treatment as well as negative perceptions among patients and caregivers are examples of risk factors for pressure on the actual possibility of providing culturally appropriate care in the hospital. [1, 4–7]

Although caregivers experience intercultural challenges on an almost daily basis [6, 8–10]; and even though the concepts of cultural competence and transcultural nursing have gained a large amount of attention within the international literature [11–16], ethical guidelines on intercultural care practices remain underexposed or lacking, leaving many care practices open to misunderstandings due to intercultural difficulties. [4, 17, 18] As such, the question of how to provide appropriate care in the context of intercultural diversity remains open to a large extent. A better understanding of the intercultural bedside care experiences of both ethnic minority patients and caregivers is crucial in finding the right directions for providing intercultural care in the hospital setting.

With this paper, we aim to fill this gap by presenting the first part (patients' perspective) of the results of a large-scale qualitative research study about the intercultural care experiences of Muslim women from Turkish and Moroccan descent, as well as of their caregivers in maternity wards in Flanders, Belgium. This focus was chosen because three fields of tension come together in this particular intercultural context.

First, a field of tension exists between the *predominant, mono-cultural biomedical approach* in the Belgian healthcare system and the multicultural and multi-religious character of the real time society. [19, 20] Although the Belgian healthcare system is officially regarded as one of the most equitable healthcare systems worldwide, it also faces an increasing diversity in its population and is continuously being challenged by the above-mentioned risk factors concerning care for ethnic minority patients. [21] Belgium has been integrating European Directives to improve the health of ethnic minority patients and some recommendations and guidelines are present in the Belgian healthcare context. [17, 19, 22, 23] Official policy obligations towards intercultural diversity in healthcare organisations, however, do not exist. [21, 22] Hence, critical reflection on the predominant biomedical approach and positive action concerning the

provision of appropriate intercultural care remains on the level of free initiative of people and organisations.

The second field of tension concerns the *culturally different understanding* in the very meaning of illness, health, treatment and care between ethnic minority patients and caregivers. As we learn from a large-scale systematic review, ethnic minority patients inevitably take their own cultural views on care with them when they are being hospitalized. [4] As such, patients' expectations, preferences, attitudes and behaviours within the care process are being influenced by culturally determined values and beliefs from their own cultural context of care. [4, 24–27] These cultural values concerning good care inevitably meet with the cultural values and beliefs of caregivers in the hospital setting. [4] The way in which these different views on health, illness and care are handled by patients and caregivers is thus an essential part of the intercultural care encounter in the hospital setting and a possible cause of misunderstandings between ethnic minority patients and caregivers.

Finally and even more challenging is the provision of qualitative care for *ethnic minority women in the maternity care setting*, since here, caregivers have to deal with a higher physical, psychological and social vulnerability of women being pregnant and giving birth within a post-migration context. [28, 29] Worldwide, immigrant women are still facing lower quality of care, higher perinatal and infant mortality, a higher risk of maternal or child health problems and barriers in access to obstetric and midwifery-led care despite the right for all women and their new-borns on quality care throughout pregnancy, childbirth and the postnatal period. [3, 30–33] Previous reviews on maternity care experiences of ethnic minority women discussed difficulties in the care relationship, difficulties in communication, the presence of racism and discrimination, the importance of family involvement and the influence of expectations and cultural practices on the women's care experiences. [27, 28, 30, 34] Due to the quick, dynamic and short-term character of maternity care, there is a lot of pressure on the quality of the intercultural care relationship between patients and their caregivers during the hospital stay.

Taken together, these three fields of tension present the complexity of providing care in the context of intercultural diversity in a challenging way. How should we understand the care relationship between ethnic minority patients and their caregivers in such a setting? How do Muslim women from Moroccan and Turkish descent experience the care during their stay in the maternity wards? Within the specific Belgian context, the focus of our research lies on the experiences of Muslim women from Turkish and Moroccan descent giving birth in maternity wards in Flanders.

## Methods

### Study design

The Grounded Theory approach was used to gain nuanced understanding of the perinatal experiences of Muslim women from Turkish and Moroccan descent. This qualitative research design suits best for exploring experiences and underlying meanings from the women's point of view. The inductive Grounded Theory approach is especially useful to understand complex phenomena and to develop a theoretical framework of these phenomena and their underlying dynamics and processes. [35] The COREQ guidelines (Consolidated Criteria for Reporting Qualitative Research) were followed to ensure rigour of the study. [36]

### Participants and sampling

Participants were included when they considered themselves as Muslim women from Turkish and Moroccan descent and were hospitalized in maternity wards (considered as a normal

setting for women to utilize during childbirth) in the province of Limburg during the last 3 years. As for the type of birth, selection was open to varying types (natural births, first and following births, caesarean, instrumental, and more complex types with specific health needs). The focus on Turkish and Moroccan descent was based on the fact that they belong to the two largest Non-European minority groups in Belgium. [37, 38] Both groups have different ethnic roots but came to Belgium as labour migrants (1960–1974) or by reunification with their families and are living in Belgium as Muslim Minorities. Furthermore, both groups are often the subject of public debate on multiculturalism and are dealing with stigmatization in the society (especially the Moroccan community). [37] Limburg was chosen because of the existence of seven coalmines and a high number of labour migrants from different nationalities working and living together since the 1920's. Since the early sixties, the mines have mainly attracted labour migrants from Turkey and Morocco. Muslim women from other origins and women who gave birth at home were excluded, as well as un-documented immigrants since their status can be linked to very specific healthcare problems. [9, 39] At first, we also excluded women who were not able to speak Dutch fluently due to extra sensitivities and risk of bias when interviewing with an interpreter.

Initially, we applied purposive sampling to recruit a rather homogenous group of first participants (6 to 8 interviews). After the analysis of these interviews, theoretical sampling was carried out based on the insights that were gathered from simultaneous analysis and the need for further clarification, variation and heterogeneity. [35] For instance, after 6 interviews it became clear that communication was a major theme. Although we initially excluded women not able to speak Dutch fluently, analysis directed us to recruit women with less language ability to clarify a new question: since communication is so important for people who are fluent in Dutch, what, then, are the experiences of people not able to speak Dutch fluently? The final size of the sample was determined by the principle of saturation, when all dimensions were identified and there was sufficient variation. [35]

Since the women were not recruited via hospital wards, several strategies were necessary to deal with expected difficulties when recruiting women from a minority group as described in Table 1. [40]

The most successful strategies were strategies with a personal connection, for instance when key persons mediated or when the interviewer regularly attended 'get-together' moments. The least successful strategy (advertising in unfamiliar social media groups) was the one that lacked direct personal contact.

**Table 1. Recruitment strategies.**

| Recruitment Strategies: | As a result: |
|---|---|
| • **Key agencies** were approached *(integration centres, schools and organizations concerning the needs of minority groups/ mothers & children* | ▪ Feasible contact persons contacted potential participants |
| • **Key persons** were contacted | ▪ Various key persons mediated between the interviewer and the Turkish/Moroccan communities |
| • **Informal 'get-together' moments** were attended in various key agencies | • Trust was built by the interviewer with 'difficult to reach participants' by repeated moments of fieldwork (*cooking & eating together*, *taking care of the children*, *discussing daily life issues*) |
| • **Snowball sampling** was performed | ▪ New participants were suggested by participants after the interview |
| • **Advertisements** were placed in a website of a key organization and in 6 social media groups | ▪ No results |

These strategies resulted in the participation of twenty-four Muslim women from Turkish (n = 11) and Moroccan (n = 13) descent in a semi-structured, in-depth interview. Included women were hospitalized in maternity care units of six different general hospitals in the province of Limburg during the past 3 years (Table 2). In these hospitals, obstetricians perform almost 95% of all deliveries and women have direct access to specialist care; consulted an obstetrician before and during pregnancy, labour and postpartum. [41, 42] As for the type of migration, the women's background varied from recently migrated women (even still during pregnancy) to women whose (grand)parents migrated to Belgium. It is important to note that the categories of first (n = 10), second (n = 13) or third generation (n = 1) are not to be understood as clear cut or well-delineated categories since their migration backgrounds reflect the tendency of transnational migration in worldwide societies according to which people migrate

**Table 2. Characteristics of the participants and the hospitals in which they gave birth.**

| Characteristics of Participants (n = 24) | | Characteristics of Hospitals (n = 6) | |
|---|---|---|---|
| **Origin:** | | **Statute of the hospital** | |
| Turkish | n = 13 | Private | n = 5 |
| Moroccan | n = 11 | Public | n = 1 |
| **Generation*:** | | **Total number of beds:** | |
| 1st generation | n = 10 | <250 | n = 2 |
| 2nd generation | n = 13 | >250 and <500 | n = 2 |
| 3th generation | n = 1 | >500 | n = 2 |
| **Delivery of youngest child:** | | **Number of beds maternity ward:** | |
| Natural delivery | n = 20 | >20 and <25 | n = 3 |
| Caesarean section | n = 4 | >25 and <50 | n = 1 |
| | | >50 (without NIC**) | n = 1 |
| **Number of children:** | | >50 (incl. NIC **) | n = 1 |
| 1 child | n = 11 | | |
| > 1 child | n = 13 | | |
| **Age of youngest child:** | | | |
| < 12 months | n = 13 | | |
| < 24 months | n = 9 | | |
| < 36 months | n = 2 | | |
| **Language of the interview:** | | | |
| Dutch | n = 20 | | |
| Dutch and English | n = 1 | | |
| Arabic | n = 3 | | |
| **Education:** | | | |
| No education | n = 3 | | |
| Secondary education | n = 12 | | |
| Technical higher education | n = 3 | | |
| University colleges | n = 4 | | |
| University | n = 2 | | |
| **Employment:** | | | |
| Unemployed | n = 15 | | |
| Parttime | n = 4 | | |
| Fulltime | n = 5 | | |

* In accordance with Timmerman [43]: migration before the age of 7 years = second generation of migration.

** NIC = Neonatal Intensive Care.

to several societies (to work, to reunite with family) during their lives. For instance, some women migrated first to another country in the European Union (e.g. neighbourhood countries) and later to Belgium. Furthermore, we will use the term "(Muslim) women" instead of the more correct term "Muslim women from Turkish and Moroccan descent" for readability reasons only.

## Data collection

Between May 2016 and December 2017, interviews were carried out by the first author (LD), a female anthropologist with experience of conducting in-depth interviews with female Muslim women in Belgium. The interviewer is a mother herself and was raised in the same region as the women from the study but is not from a Turkish or Moroccan origin herself. The interviewer was aware of the fact that taking time and following the pace of the women, was the most important prerequisite to carry out meaningful interviews. As such, the interviewer not only spent time before and after each interview to engage within the daily life of the woman and her family but also allowed her to set the pace during the interview itself. As such, the interviewer did not easily interrupt elaborations by the women during the interview.

Depending on the place where the women felt most comfortable, interviews took place either in their own home (n = 20), their parents' home (n = 1) or in a private room within a familiar organisation (n = 3). Interviews lasted on average 115 minutes (range 50'-225') and were conducted in the women's preferred language, most of them in Dutch (n = 20). One interview took place mostly in Dutch and was supported by English clarifications, although we have to specify that this mother could not express herself in Dutch during her hospitalization (n = 1). In one case, understanding one another turned out to be more difficult than expected because the woman's language ability was sufficient for everyday situations but not to express experiences or profound feelings. In this case, the interviewer returned for an additional interview with an interpreter to deepen the data. In total, three interviews took place by the interviewer accompanied by an Arabic interpreter (formal (n = 1), informal (n = 2)). Bearing the sensitivity of the subject in mind, it was important that the women themselves could choose an interpreter with whom they felt most comfortable. Afterwards, an independent interpreter (also a qualitative researcher herself) checked the transcriptions of these interviews for accuracy of translation.

The semi-structured interviews followed an interview-guide based on a previously published systematic literature review [4]. The interview guide was adapted after a pilot interview and refined after discussions within the research team and a meeting with other experts in the field. The open and non-suggestive questions allowed the women to describe their experiences freely while holding a focus on their experiences [35]. Each interview started with a personal introduction of the interviewer with focus on the shared experience of being a mother. After this, the women were asked: 'Can you tell me all about your delivery, from one mother to another?' The interview-guide was used in a flexible manner and proved to be very helpful to check if all the important themes were covered throughout the interview (S1 Appendix). At the end of the interviews, the women were asked about what they experienced as the most important themes discussed and whether there was something else important to discuss that had not been elaborated yet. The themes in need of more attention were noted in the margins of the interview-guide after each interview.

## Ethical considerations

In March 2016, Ethical approval (G 2016 03 531) was obtained by the Social and Societal Ethics Committee (SMEC) of KU Leuven. All the participating women received a written

information brochure about the study purpose and nature, information of the research group, procedure, the rights as study participants and contact details. Due to the sensitivity of the subject and/or presence of an interpreter, additional ethical considerations were crucial throughout the interview process (S2 Appendix).

All the women agreed to digital recording. During the first interview, the women were asked permission to contact them afterwards in case of additional questions. In one case, the interviewer returned for an additional interview to complete the data and in two cases, the interviewer asked additional questions by phone. Interviews were never interrupted or stopped once they had started although some interviews (n = 5) were rescheduled last minute due to illness, doctor appointments, etc. One women withdrew due to work-related difficulties.

## Data analysis

The interviewer (LD) made field notes and a narrative report directly after each interview. Interviews were meticulously transcribed (LD) *ad verbatim* (non-verbal signs included) as soon as possible after the interview. We used the Qualitative Analysis Guide of Leuven (QUA-GOL) [44], to analyse the data in accordance with the Grounded Theory approach. It consisted of two parts, a preparatory part (pen and paper stage) and the actual coding of the data (by the software QRS international's Nvivo 11, 2016). [44]

During the preparatory stage, two researchers (LD and YD) carefully (re)read the transcriptions and descriptive reports and marked significant events, facts or meaningful elements. LD and YD independently made a conceptual scheme for each interview to capture the essence and to cluster concrete experiences in patterns or themes on a more abstract conceptual level. Thus, meaningful themes grounded in the data were discovered rather than breaking down the data as a result of a line-by line coding process. YD and LD compared and discussed these conceptual schemes for similarities or discrepancies. The other supervisors of the multidisciplinary research team (CG and BDdC) simultaneously read seven of the richest interviews. During regular meetings, we discussed the identified patterns, potential discrepancies and themes that needed more clarification. After ten conceptual schemes, we developed an overarching scheme in which key themes were compared for similarities and differences. After this, the overarching scheme was continuously tested for appropriateness by following and previous interviews. From this stage on, a constant forward-backward movement occurred between analysis in one interview and across interviews and between analysis on a basic level and on a higher level of conceptualization. After this, a peer debriefing took place (March 2018) in which the interpretation of the transcripts and results were thoroughly assessed and discussed by an interdisciplinary panel of external experts.

Based on the themes from the preparatory stage, LD made a common list of analytically meaningful codes in preparation of the coding process in Nvivo 11. During the stage of the actual coding process in Nvivo 11, LD linked all the relevant text fragments to appropriate codes by using the list of codes identified in the preparatory stage. This was followed by a close examination of the meaning, dimensions and characteristics of the concepts by means of their associated quotations. This helped us to extract the essential structure and to develop a theoretical framework as an answer to the research question. We verified the accuracy of this framework by means of constant comparison with all the individual interviews and conceptual schemes. Finally, we translated the findings into a narrative storyline and illustrated this by relevant translated quotations. The multidisciplinary research team regularly checked and discussed the actual coding process, the development of the concepts and the framework.

## Results

The narratives of Muslim women from Turkish and Moroccan descent present the care experiences in a Flemish maternity ward as a dynamic and long-term care process, which starts from the prenatal consultations and lasts until the postnatal check-up. During this perinatal care process, the women were engaged in various kinds of relationships with multiple caregivers, each influencing the women's care experiences in their own way. From the narratives we learn that the way in which the women ultimately experienced the full care process was determined by the dynamic interplay and interrelatedness of two essential dimensions, viz. *Ereignis* and *Erlebnis*. A German translation of these concepts was necessary due to the lack of a well-grounded translation from Dutch to English for the terms *['Gebeuren' i.e. 'Ereignis']* and *['Beleving' i.e. 'Erlebnis']*. The correct translation was frequently discussed, not only with the multidisciplinary research group but also with several experts in the field.

The first dimension, *Ereignis*, refers to the women's experiences of '*what*' actually happened during the care process. It entails the women's experiences of the accuracy or attentiveness of caregivers when performing a variety of care interventions (medical, technical acts). It represents the women's experiences of what caregivers did to take care of (or even to cure) them when giving birth. In essence, it refers to the question: "*What attention did various caregivers have for the acts that happened or needed to happen*?" as seen through the eyes and experiences of the women.

The second dimension, *Erlebnis*, entails the women's experiences of '*how*' or '*the way in which*' the care process took place. Here, the women referred to the caregivers' attentiveness for their emotions, feelings and wishes as well as to their various 'beings', as a person, a patient, a new mother, a daughter, a wife and as a Muslim women. Essentially, this dimension refers to the question: "*What attention did various caregivers have for 'the way in which' these things happened to me*?" as experienced by the women?"

These two dimensions are *interrelated* and the *dynamic interplay* between them determined how the Muslim women experienced the care process. Whether or not they were in balance depended on the two-dimensional attention that was given to them by the various caregivers across the whole care process.

On a more fundamental level, the narratives revealed that both dimensions of the care process as well as all the care interactions between the women and the caregivers are embedded in the women's own *cultural background*, by which they give meaning to their care experiences. This means that the women's cultural context cannot be understood as a static set of values, preferences or traditions but has to be understood as a dynamic process in which a symbolic, emotional or moral meaning was given to the women's experiences *throughout* and *within* the process of caregiving and care-receiving. In this relational process, the women balance between their own cultural system of values, beliefs and traditions and the actual reality of values, beliefs and traditions of the multiple caregivers in the hospital. As such, different (cultural) views on the childbearing body, on appropriate treatment and good care come together *during* and *across* many different care interactions between the women and their caregivers.

Fig 1 provides an illustrative overview of these dimensions and the determinants of the women's care experiences. In the following sections, we will first describe the interrelatedness of the two essential dimensions of *Ereignis* and *Erlebnis*. The second part presents the influence of culture as a *meaning system* affecting the women's experiences in a fundamental way.

### Four ways of interrelatedness

The women's narratives showed that the dynamic interplay between the two essential dimensions (*Ereignis* and *Erlebnis*) and the way in which both dimensions were interrelated

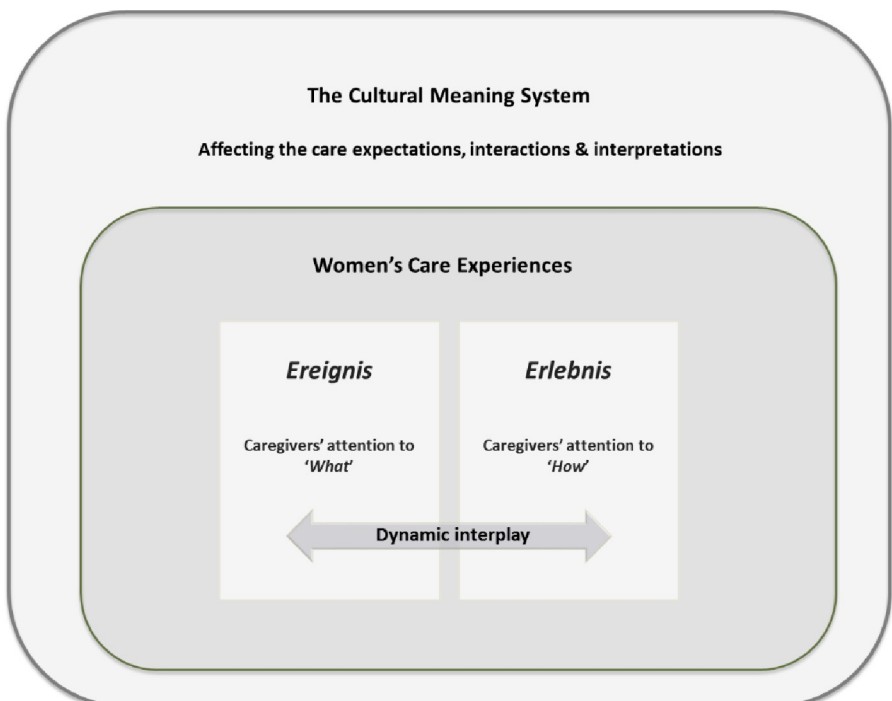

**Fig 1. Theoretical framework.**

determine the outcome of the perinatal care process. Four ways of interrelatedness between the two essential dimensions define the outcome of the long-term care process as experienced by the women. The four outcomes are summarized in Fig 2.

**Protocolized care.** The first outcome can be presented as a care experience in which there was a fair amount of attention of the various caregivers to the dimension of *Ereigni*s and little or no attention to the dimension of *Erlebnis*. In the women's experiences, technical care was performed based on strict protocols and procedures. By this, the women experienced that everything went well on the level of the medical outcome. In general, however, they perceived little attention to their experiences, feelings or emotions. Although everything went well in a medical and technical sense, the care process did not happen as the women wished for. On the contrary, they felt forced by various caregivers to follow rigorous rules in a quick pace, while they felt as if there was no regard for their own voice and choices in the care process. In some cases, the women were even told to 'cut the comedy and just follow the procedures'.

Altogether, the women experienced that not many reciprocal care relationships existed here since they felt that most caregivers did not communicate well and were often 'too busy to care'. Nevertheless, caregivers handled with medically competent, technical 'cure' (e.g. when answering complications) despite the restricted room for individual, religious or cultural wishes of the women. Due to this, the women only felt welcome as a patient but not as a person.

*"I did not know what was happening to me [. . .] In the delivery room, I suddenly started to cry very hard because they said 'take off those rags, lie down and don't move' so there was nothing soothing left at all. I wanted to light a candle and let the Quran play a bit but that was not allowed, everything had to be fast and quick. Everything was very compelling, I just had to listen and follow and I could not be the director of my own childbearing. [. . .] Also in the morning, they invaded [in my room] just like that. While you are still half-asleep, they*

| Ereignis | Erlebnis | Outcome |
|---|---|---|
| + | - | **Protocolized care**. Emphasis on procedures. Recognized as a patient but not as a human being. Care went well although it was not in accordance with the women's wishes. No self-agency. |
| - | - | **Uncaring care.** Inadequate medical performance. No recognition of their role as a patient, nor as a human being. Overwhelmed by powerlessness. No self-agency. |
| + | + | **Embraced care**. Women felt welcome. Full recognition of their role as a patient and as a human being. High self-agency. |
| - | + | **Ambiguous care**. Resignation by the women towards medical inadequacy. Partial recognition of their role as patient or as a human being. More or less self-agency (varying from different care relationships or specific wards) |

**Fig 2. Four ways of interrelatedness.**

*came to measure and weight her and then they continued like this, there was no pace in it and I could not rest after the birth [. . .] no, those were just standard procedures that must be carried out in the hospital."* (6)

**Uncaring care.**   The second way of interrelatedness was revealed by the women who felt overcome by their experiences since the various caregivers paid little or no attention to both dimensions *Ereignis* and *Erlebnis*. On top of the fact that the women's own voice and feelings were not acknowledged (cfr. *protocolized care*), technical and medical inadequacy occurred. In these narratives, various caregivers did not recognize the seriousness of physical and emotional difficulties as expressed by the women themselves:

*"They did not believe me and said*: '*Madam, you should not make it bigger nor turn it into a drama, those are just a few drops [of blood] you know. The baby is coming and within 48 hours you will have given birth'. What 48 hours*?! *Within 45 minutes, I had my baby, delivered by myself [without caregivers around] because they did not believe me."* (9)

Many women mentioned that 'something was wrong' but caregivers failed to recognize the seriousness of their signals. As a result, complications occurred and care became urgent and acute. A care process started within which many caregivers were immediately and overwhelmingly present in a medical and technical sense, but nobody noticed the patient as a person. As one woman described the experience of her acute caesarean section in which 'inhumane' caregivers did not explain the things they were doing nor were they talking to her to reassure her:

*"I keep saying, with my first [caesarean section] it happened just like a sort of . . . yes, a sort of slaughter."* (21)

In such cases, little or no reciprocal care relationships existed and the women felt lonely due to caregivers being indifferent, unhelpful or uncaring. Some felt that caregivers treated them in a brutal, annoyed, harsh or irritated manner. Overall, they mentioned a lack of attentiveness not only towards their physical needs (e.g. no check-up) but also towards their emotional and psychological needs. Caregivers only 'did' things quickly (even very intrusively), but did not interact with the women nor explained what was going on or what was going to happen.

*"Suddenly many caregivers came in and out and I was thinking 'I don't have a clue who you are or what you are going to do'. I was just lying there with my legs wide open and they were just doing stuff. I did not understand it at all. [. . .] It was annoying, of course, because maybe, I even wouldn't have wanted this but they would have done it anyway. I didn't even know until later, when my husband said 'Look, they are sewing' I said 'Sewing?' 'Yes, they had to cut you' and I said 'Cut me?' I was thinking 'Oh my God, what is going on? What are you doing down there?!' Yes, that is really scary."* (14)

These women expressed feelings of powerlessness, of being at the mercy of their caregivers and thus, a lack of self-agency existed. Many of them expressed that they felt treated as a number instead of as a human person since their concerns, questions and worries remained largely unseen.

**Embraced care.** The third way of interrelatedness of the two dimensions was revealed in the women's experiences where a lot of attention to both dimensions *Ereignis* and *Erlebnis* existed. Various caregivers frequently and spontaneously asked them about their concerns, questions or worries. There was personal contact and meaningfulness within the various care interactions and caregivers were helpful, attentive and kind to them.

*"They were always very friendly, helpful and asked by their selves 'Can we do this or that for you?' So they suggested things themselves, which made me very reassured and made me think 'Look, they are really helping me, not just in a curtly way'."* (2)

According to these women, caregivers emphasized that taking care of their needs was not an effort to them and that they loved even the less pleasant parts of the job. Here, complications also happened but were handled differently than in the *uncaring care* narratives. Competent care was important to the women but instead of a sole focus to the medical and technical aspects of care, they were surrounded by extra care, respectful attitudes of caregivers and were talked and guided through difficult moments. The women felt that they were taken seriously. They did not feel lonely or ignored in their own knowledge, fears or worries. On the contrary, they felt embraced as a human person and did not feel like a number (versus *uncaring* and *protocolized care*). The women described the various ways in which care was adapted to their own specific needs. They felt in control of the care process.

*"I honestly expected them to be much stricter and that I would have to listen to them but I was actually in charge. [. . .] They were just guiding me in what I want and did not act like: 'No, mama!' No, they just asked: 'What do you want?' At least, you have an opinion and not like: 'You are going to do this because we are in charge.'"* (22)

The women felt recognized as a person with room for their own personal, religious and cultural wishes.

**Ambiguous care.** The fourth type evolved out of the ambiguity in some care experiences. One reason for this ambiguity was due to the presence of (some) medical inadequacy (*Ereignis* dimension) that contradicted with the presence of attentiveness towards the *Erlebnis* dimension, either by the same or by other caregivers. The women forgave and resigned to medical inadequacy when (some of) the caregivers paid extra attention to the way in which the women felt during the care process even though it was hard or (some) things went wrong. Some narratives displayed ambiguous feelings mainly due to the overall good intentions of the various caregivers despite negative feelings caused by medical complications and inadequacies. In these narratives, complications occurred but caregivers were at least trying to give good care by listening to them about how they felt and by answering to one's wishes. In other narratives, ambiguity existed when some of the caregivers were inattentive to either of the two dimensions while other caregivers (e.g. in other shifts or wards) showed a (very) high attention to at least the *Erlebnis* dimension.

> *"I have always had a hard time when caregivers check down under, so I asked my gynaecologist 'Can you please be careful?' She really took that into account and reassured me [. . .] She wasn't too direct like 'Open your legs and I will take a look.' No, she was like 'Take your time, I'll wait, just relax.' [. . .] While [another caregiver] came in and said: 'Your husband can't help it either, it is painful, you're going to have pain and you'll have to hold on.' [. . .] She said: 'I'm going to have a look.' And then I asked again: 'Can you please do it calmly?' But she did not. She "looked"! That was the difference, she did it very directly, but I was not used to that. (15)*

Here, the presence of caregivers who were especially attentive to the *Erlebnis* level, contrasted with the presence of caregivers without attention to *Erlebnis* during the same hospitalization. When the women met attentive caregivers after previous negative events (e.g. maternity ward versus obstetric rooms), they regained self-agency and recognition and resigned themselves to previously experienced negative events. For example, when the baby had to be admitted to the Neonatal Intensive Care Unit (NICU), the sharpness of the events that happened during delivery (even very intrusive moments) were toned down by most of the women. Altogether, these women felt largely positive about the care process despite previous negative experiences, which was subscribed to the extremely caring attitudes and the cautious medical way in which caregivers in the NICU took care for them and their new-borns.

## The women's care experiences

Based on the women's narratives, we detected that they all, in their own unique way, showed one pivotal way of interrelatedness across the various care relationships. From an overarching view, their own care experiences were either predominantly *protocolized*, or *uncaring*, or *embraced* or *ambiguous*. As such, the women described various similar moments of interactions throughout the care process in general. Of course it did happen that some particular care relationships were different from their overarching experiences, but in such cases, these events had a smaller influence on the women's care experience in general. For instance, one woman described an overarching '*uncaring*' process but encountered one particular, very '*embraced*' caring nursing student. This specific encounter with a caring student, however, less significantly influenced the overall picture of that woman's experience.

In general, the women's experiences revealed that the attention of caregivers to '*how'* they felt during their care and '*the way in which'* care was carried out (*Erlebnis*) was at least equally

important to the caregivers' attention of '*what*' happened and the sort of (in)competent care caregivers performed (*Ereignis*).

> "*Introduce yourself and be friendly. [. . .] I will feel reassured a bit if you say*: '*I will come to you, I will help you and we will get through it together. Just come on.*' *That is completely different from just invading, doing what you want to do and then just leave, by which I think*: '*Who are you and what have you actually done*? *What are you going to do*?' *Yes, that is really, really different.*" (14)

Notably, the narratives also showed that '*the way in which*' caregivers took care for the women and their new-borns determined the women's fundamental care experience to a high degree.

> "*In the hospital, they say things very quickly and then they move on. Whereas if they would sit down with you and hold your hand just a little while. Because they are touching your breasts anyway, so why can't they also hold my hand*?" (4)

## Culture as a meaning system

The women's care experiences, i.e. the four ways of interrelatedness and the respective care interactions between caregivers and women, were fundamentally embedded within the women's own cultural meaning system. This cultural meaning system was revealed as a dynamic relational process by which the women's care experiences and their care interactions with caregivers took on an emotional, symbolic and moral meaning. As mentioned previously, the women's cultural context of values, beliefs, practices and traditions gave meaning to their expectations, preferences, attitudes and behaviours regarding the care process. Remarkably, most women did not *separately* mention culture or particular cultural practices or traditions as being *explicitly* important for their overall care experience. Nevertheless, the narratives revealed that the women's culture was present as a fundamental meaning system that affected every aspect of their care experience in the hospital. The way in which this actually happened, or the intensity of its importance, differed profoundly throughout the narratives. Every woman had a unique way of describing the actual influence of culture on their care experiences.

Notwithstanding these individual varieties, they also all shared the influence of culture in at least three substantive aspects of the women's overall care experiences. Their cultural system affected their *expectations* about care and care relationships, it affected what happened during the *care interactions* in the hospital and it influenced the way in which women *interpreted and coped with* the emotional event of giving birth.

**Culture affecting the women's expectations.** Most women indicated that they did not have *explicit* expectations towards the care experiences in the hospital during the process of giving birth. As such, they expressed thoughts like: "Come what may" or "I do not know what to expect" or "It is in Gods' hands". On closer look, though, we detected that all the narratives revealed *implicit* expectations and *underlying* preconceptions that were mostly self-evident within the women's own cultural meaning system, but which turned out to be less obvious within the actual reality of the hospital culture. For instance, one of the women pointed at the difference between the cultural meaning of giving birth (as something that is highly special) and the meaning of giving birth within the hospital culture (as something that is quite normal).

> "*In our culture, we really say that when you are giving birth, you are literally standing with one foot in your grave. Really, that's how hard they see it. It's quite something.*" (22)

Even more, the women's care experiences were often determined by the manner in which their unique expectations about the delivery and care process (*viz*. 'how care should be') came true within the hospital reality. Some women struggled when their expectations did not come true due to differences in views with caregivers on the special status of a childbearing woman, the desired way of giving birth or the sort of caregivers' support. For instance, many women expected to get extra support by the caregivers as a result of their cultural preconceptions about the vulnerable and emotional position of a childbearing woman and their emphasis on staying strong during a natural birth (since pain was, in their view, a test from God).

> *"I want to be cared for with attentiveness [. . .] and not like 'Just wait your turn and then give birth.' They really have to listen to you as a person and to ask about your feelings especially with pregnant woman. [. . .] that moment, that is something special and you certainly need being supported and motivated by saying that you are going to do it well and that they are here for you if something goes wrong. So that they are really present by listening to you."* (15)

The expectation of being surrounded by attentive and supportive caregivers was challenged when the women only faced a rushed presence of caregivers (e.g. *uncaring* or *protocolized* care) or when being pregnant or giving birth were considered by caregivers as 'normal' and not as something very special. For instance, when caregivers solely focused on the baby's care without sufficiently acknowledging the mothers' needs.

> *"Some [caregivers] only care for the baby but the mother needs a little bit of care too, we also have symptoms and pain. This time, they actually looked at us, not just at the child. That was nice. If we feel very bad, how can we take care of the child? [. . .] So I think they should give more attention to the mother."* (13)

Nevertheless, when complications occurred by which the delivery did not turn out the way they expected it to be (e.g. epidural, cesarean section), most women did not assess it as being insurmountable when the care process happened *together* with caregivers. Many women expected a warm and attentive care process with appreciation for the special character of giving birth, during which a reciprocal care relationship with caregivers, characterized by *mutual kindness and equality*, could be realized. The women expressed their expectation of 'being treated in the same way as you treat others' (e.g. mutual kindness) as something that is important within their culture (e.g. '*be kind to others and then you will receive kindness in return*'). The care relationship became seriously under pressure when caregivers were harsh or brutal while the women themselves were trying to be as polite as could be. The expectation of reciprocal care was also expressed in the women's expectations of realizing care together, in a relation of equality. As such, they also expected the caregivers to help them with taking up their own role:

> *"I do not know, I expected when I was entering [the hospital] that they would say first: 'How do you want to give birth?' I believed that this was going to be the first thing they would say, and from there on, that a midwife was constantly going to be with me."* (6)

Furthermore, most of the women's present expectations were influenced by previous care interactions and the way in which they coped with these previous events according to their cultural meaning system. Some women felt more confident due to these previous experiences (e.g. with supportive caregivers) while others tried to cope with previously insensitive caregivers (e.g. towards themselves as a patient or as a Muslim woman) and previous negative care

experiences (e.g. during traumatic deliveries, miscarriages or prenatal hospitalizations). Current difficulties were aggravated as soon as caregivers reacted in a similar way as before (e.g. by not taking them seriously in their needs and worries, again. . .).

> *"After the blood loss we went back to the emergency, we ring the bell and that woman said: "Ah, could you not persevere at home?" So what did she do? Once again, she did not check below." (15)*

Previous deliveries in foreign countries also influenced the women's expectations since they expected the same care as they were used to have (e.g. Netherlands, Turkey or Spain) or even expected better care (than for instance in Morocco). However, this expectation did not come true when women were confronted with a caregivers' lack of attention especially to their *Erlebnis (uncaring, protocolized care)*.

Another influence on the women's expectations were the shared narratives from family members and friends, intensified by the use of social media and internet (by which the women also stayed connected with their family and friends worldwide). By these, the women explored every manner of 'how care can or should be', prepared themselves to the moment of delivery or braced themselves due to horror stories coming to them from almost everywhere (e.g. failed epidural anesthesia, intrusive caesarean sections or racist caregivers).

In general, the lack of congruence between the women's expectations and the reality in the hospital sometimes led to far-reaching consequences.

> *"I did not expect that care would be so serious here, that was really a shock and because of this I'm really thinking of: 'Well, I'm not taking another child yet, not for now.'" (14)*

This incongruence sometimes caused women to question their choice of giving birth in Belgium or even their own migration:

> *"When that first nurse treated me so [badly], I thought: 'Oh no, it is going to be like in Morocco.' [. . .] I have left my whole family behind, my parents, everyone, the whole family. . . to have it better, not for having the same thing and then [I] was thinking: 'Is this better?' (8)*

From the women's narratives, we also learned that prenatal check-ups had a great impact on the women's care experiences, since they created the opportunity to talk through the women's expectations, previous difficulties or worries and to prevent misunderstandings due to different views about the care process. This was especially important when the women described a significant difference in meaning of being pregnant and giving birth between their own culture and the (biomedical) culture of the hospital.

**Culture affecting the care interactions.** The women's care experiences were also shaped by the way in which their own cultural meaning system intermingled with the caregivers' cultural context during the care process. The narratives showed that most values, practices, beliefs or preferences embedded in the women's culture were not posing insurmountable challenges to the care interactions *if and only if* the caregivers were generally open and attentive to the women's needs, worries, wishes and feelings. That is, when they had well-balanced attention for both the *Ereignis* and *Erlebnis* dimension of the care process (e.g. embraced care). In these situations, the women felt welcome and recognized as a person with her own specific cultural context, which could be part of the care process, and the care interactions with their caregivers.

For example, when caregivers were generally attentive, the women easily found practical ways to integrate some of their *cultural and religious traditions* within the care process, like for instance listening to the Quran during delivery, whispering the 'Adhan' (first call to prayer) or ''Shahâda' (profession of faith) in the child's ear, rubbing the baby's palate with a piece of date, swaddling the baby in and eating halal food. Most practices happened implicitly, almost silently (like the 'Adhan' or the date), or when the caregivers were absent. Nevertheless, most women really felt respected when the caregivers reacted in a positive or neutral way to the presence and integration of cultural practices into the care process. For example, the women felt respected as a human being and as a new parent when caregivers paused the care acts for a few minutes especially to give the parents the chance to carry out the 'Shahâda' in silence.

> *"They really respected that, they stopped the things they were doing because we had to whisper this as soon as the baby comes out [. . .] That's one of the first things she should hear and then she is actually born as a Muslim [. . .] That's her 'Shahâda' we say, it is very important that it is done. And they [the caregivers] acted as if they already knew that. I think they have seen that before. They just had a smile on their face. It was not like: 'What's he doing there?'" (22)*

When caregivers, on the contrary, were not attentive or open to the *Erlebnis* dimension of care, the women experienced less room for their cultural values, traditions and preferences, and sometimes even had to deal with negative reactions to it. Consequently, it happened that already existing difficulties in the care relationship further intensified and added up. For instance, when caregivers expressed themselves contemptuously when the women wanted to listen to the Quran during delivery, and when this happened *on top of* other uncaring attitudes (not only towards their culture), there was nothing soothing or reassuring left for these women.

> *"[Then I thought to myself]: 'If I want to listen to the Quran, I will do that. Why do you react this way [i.e. contemptuously]? You do not have to find it right. You do not have to find it correct. But please keep your opinion to yourself. Do not show what you think of it!' It really was not the moment. Outside [in another context], he [gynaecologist] can talk to me about it, but not while I'm giving birth! I'm not saying that he should respect it, that he should think it is a good idea. No. Maybe according to him it is totally useless, I can agree with that. But why react so negatively? [. . .] You [as a caregiver] must be able to reassure someone who is giving birth. You should not give her a negative feeling about it [her delivery]." (5)*

When there was no room for the implicit presence of culture within the care process, differences between the two cultural meaning systems (i.e. between patients and caregivers/hospital) could cause misunderstandings in the care interactions and put the care relationship under pressure. In some cases, these differences have led to underlying feelings of cultural prejudice, discrimination or racism throughout the care experiences of the women, although only a few explicitly identified discrimination or racism. As one women, for instance, pronounced underlying cultural prejudices of her caregivers.

> *"When I gave her a bath and dressed her, the nurse asked 'Oh, what is the reason that you already knew it that well?' She immediately asked 'Do you have a lot of brothers and sisters?' [. . .] I said: 'No, I do not'. She assumed right away that I have a huge amount of brothers and sisters [. . .] and I answered: 'No, I have been studying to become a caregiver.' 'Ah, oh, oh, oh, ok.' And it really looked as if she was shocked. [. . .] That was so strange. But at some point you start to laugh and then you think: 'Oh well, what difference does it make?'" (14)*

Other women stipulated that according to them, there was no racism or discrimination in the hospital since caregivers 'are obliged to treat all patients in the same way and are not allowed to discriminate'. Remarkable, however, is the fact that the *absence* of discrimination was regularly expressed explicitly, while the *presence* of racism or discrimination was mostly expressed in a somewhat covered way.

Particularly important for the care interactions was the way in which caregivers reacted to the role of *family members* as informal caregivers (e.g. their practical and emotional support was described as being important within the women's culture). Recognition of their role was crucial for the care interaction between women and caregivers. Some caregivers failed to recognize the family as a companion in the women's care or even treated them as a burden when they applied the visitation regulations very rigidly, or when they did not communicate well with the family about their various roles as informal caregivers, surrounding the woman.

*"I can choose one person, but I have different kinds of support from each person. My boyfriend might have given me patience and rest. My daughter gave me love, and my mother would have set me at ease. You get something different from everyone. They [the caregivers] have to understand that, you know?" (9)*

From the women's narratives, we learned that difficulties in the care interactions could be aggravated due to the manner in which caregivers sent visitors away. This became especially difficult when the mothers already felt lonely due to the caregivers' absence, as happened in the case of *protocolized* or *uncaring care* (cf. supra). Some women expressed that they experienced a fear that caregivers would no longer care for them due to discussions with their family members. On the other hand, the presence and role of the family was sometimes also experienced as overwhelming by the women themselves, because of the families' overwhelming presence (also via social media), multiple interferences, incompatible advices, or crowd visitations. Caregivers who sensed and recognized the women's discomfort because of an overwhelming family presence, and who mediated in a sensitive way, made the women feel understood, respected and cared for.

*"Nobody could hear it, but she whispered very quietly: 'Do you want me to send these people outside?' [. . .] I could not say it to them myself, that is impossible. Because these people came for me with good intentions. So I was so relieved [..]. She understood that I was having a hard time." (15)*

The absence of close family due to the women's migration was also a difficult experience for some women, which became ameliorated when caregivers in some way took over the role of absent family.

*"I cried sometimes because my mother died and my father is still living over there. At some point, they [nurses] said: 'I'm here for you' or 'How are you? Do you want something?' [. . .] Especially when you have nobody, that feeling is just incredible. (20)*

Overall, the women appreciated caregivers even more when they showed a genuine interest in them, or started a conversation about their social environment, origin or culture in an open way and recognized their cultural context as an important dimension of the women as a person. In such cases, the care relationship integrated both the *Ereignis* and *Erlebnis* dimension in a way that the cultural meaning system of the women involved could be an essential part of the whole experience.

**Culture affecting the interpretation and coping of the experiences.** Culture did not only affect the women's expectations and their care interactions but also the women's interpretation of why things happened in the way they did. When the delivery and care process went well, the women subscribed the good outcome to their own share of being polite, self-reliant, communicative and non-judgmental. Just like we detected in the part on the women's expectations (cf. supra), mutual kindness and equality ('*being treated in the same way as you treat others*') is something that the women expressed as an important religious and cultural belief (e.g. '*be kind to others and then you will receive kindness in return*'). As such, a good care process was being interpreted as something 'they 'deserved' because they were empathic and kind to the caregivers, and did not bother them unnecessary.

On the contrary, when the care process went rough, women tried to understand why things turned out this way and tried to give meaning to the question 'Why did the caregivers act in the way they did?' Also here, the women's cultural context was part of their interpretation. Metaphorically speaking, the '*ghost of discrimination*' was present in many narratives as a more covered form of discrimination or racism. This '*ghost of discrimination*' became visible when women asked themselves whether or not a certain act or omission had something to do with them being a Muslim person, their own Turkish or Moroccan origin, wearing the veil or a (perceived) lack of language efficacy: 'Did the caregiver react this way because of this or that, or because I am a Muslim women?' For example, when caregivers reacted in a non-communicative or unfriendly way, women asked themselves: 'Is the caregiver reacting strange due to their character, age or working experience or is it because of my veil?' They asked themselves whether the difficulties they were dealing with in certain cases were the same for other people or, on the contrary, a matter of discrimination.

> "*This midwife was rude to me, I do not know if it was because of my veil, but it was the way in which she said: 'Take off these rags'. She could have said just as well: 'Take of your veil' in a more polite way but when she [also] loosed her temper with a nursing student, I thought something like . . . in the beginning I thought to myself: 'Yes, she is a racist'. You immediately have the tendency to judge like: 'She is a racist' but when I saw her being infuriated at the student, I thought: 'Maybe, brutality is just part of her nature' and she was an older women, at least 50 years old, and when they grow older they are less pleasant to deal with since they are less patient.*" (5)

For example, some women asked themselves why caregivers were spending more time with other patients.

> "*When I went for check-ups, I noticed that everyone goes in and really was staying there for half an hour, but when I went in, I only had to stay for only five minutes, so what is the difference now? Is there something about me? Or about her?*" (23)

In other narratives, women described this '*ghost of discrimination*' as something that happens non-verbally, as something that 'you just feel', when 'you feel less [worthy]' or when 'they looked at me differently'.

> "*I had the feeling that he was somehow contemptuous, like: 'Okay, that is a Turkish woman, who just came to give birth to another of her, I do not know, umpteenth child.' I don't know what he was thinking about me, but I didn't like it. Look, I was born here too. I also have studied here and I deserve the same amount of attention as anyone else.*" (5)

A particular component of this experience is related to some of the women's religious and cultural practice of *wearing a veil*. Some caregivers did not react to it and thus implicitly reassured the women that within the hospital, there was no discrimination between people because of differences in cultural background (a form of discrimination that they regularly experienced in the broader society). Other caregivers reacted in a strange or less positive way by which some women experienced that caregivers did seem to estimate them negatively because of their veil.

> *"It is reassuring [when they do not react to the veil] because it is not nice. You always have to prove yourself twice as a Muslim woman. I always find it annoying." (5)*

Nevertheless, it was remarkable that most women expressed their appreciation for the care they received, even when (very) negative events happened. Most women showed a positive way of coping with negative events (e.g. a predominantly resignation towards negative events). The reason why the women coped with negative events in this way, was not always entirely clear and based on a combination of several reasons (although culture influenced this positive way of coping, it was not only caused by cultural values, beliefs or cultural interpretation of the events). One of the reasons showed itself in the fact that all ended well since all the mothers and newborns survived. Thus, even when the women dealt with *uncaring care*, most of them eventually came to terms with it since the medical outcome was more or less good. Another reason for the women's positive way of coping was their faith in Allah and their gratefulness for every outcome that Allah gave. These women expressed that '*everything happens for a reason*' and that '*Allah will not give you something you cannot handle*'. Some complications were interpreted as a life lesson and an opportunity for personal growth, assigned to them by Allah. Against this background, women sometimes minimized the severity of complications (e.g. when caregivers discovered a congenital defect when it was too late for an abortion). The women did not feel strongly about this since these children were a gift from God and they would never terminate the pregnancy for it.

> *"Of course, you are just frightened because it is about your children. You are a bit shocked if something is wrong. I had to cry but at a certain point I thought . . . yes, you know we believe in God, some people don't, but we do believe and I thought: 'God doesn't give you something that you cannot handle, so be happy, there are still people with things that are worse.' You always have to be a little bit positive." (20)*

Still another reason for this predominant positive way of coping despite negative events was, as we mentioned before, that some of the difficulties were countered by the very caring attitudes and the attentiveness of caregivers towards their *Erlebnis (viz. ambiguous care)*.

The opposite happened when difficulties, inadequacy in care, lack of attention to either of both dimensions (viz. *uncaring*, *protocolized*, *ambiguous* care) and caregivers' negative reactions towards the women's cultural context caused the women to feel anxious, distressed, mournful, disappointed, lonely and angry. One woman even wished to be dead due to incomprehensible caregivers (*uncaring* care). Here, women felt abandoned and unwelcome by which they reacted by giving up or withdrawing oneself from the care relationship or even from care in general (e.g. by leaving the hospital early or by expressing the intention of not returning back to the hospital). Some women felt it was useless to address their complaint about mistakes to the designated department. They no longer trusted the possibility of a good outcome. In such situations, the overall result was a full disconnection from meaningful care or care

relationships. Most narratives, however, emphasized that either way, the women would never forget the care performed during the emotional event of giving birth.

## Discussion

### Women, care and culture

This study shows that the women's principal focus in describing their perinatal care experiences was on the *care interactions* with their caregivers and on the two-dimensional attention of caregivers towards '*Ereignis'* and '*Erlebnis*' rather than a caregivers' lack of attention towards cultural questions during the care process. As such, our study is consistent with existing studies in which the quality of intercultural care depends in the first place on the nature and the quality of the care relationships between the ethnic minority patients and their caregivers rather than on the way in which patients and caregivers handle cultural differences during the care process. [4, 28, 45]

Despite the gratitude of most women towards the safe care outcome, they did not feel satisfied with their perinatal care process when caregivers did not have enough attention to the things that needed to happen (*Ereignis*), or when they did not have enough attention to the *way in which* they felt during the care process (*Erlebnis*). Tensions in the care relationships with caregivers (e.g. due to difficulties in communication, the caregivers' negative attitudes or their lack of support) were intensified by the lack of room for the women's cultural context.

Due to the emphasis by the women on the care interactions with caregivers, we could ask ourselves if we need to adjust maternity care in this intercultural context or whether it would be sufficient to focus on the quality of the care relationships between these women and their caregivers? Is it sufficient to start from a person-centred approach to enhance the quality of intercultural care or do we have to take the women's cultural context more explicitly into account?

**The importance of the relational care process.**   As defined by the WHO a person-centred, safe, effective, timely, efficient and equitable care is important to improve the desired health outcome and the quality of healthcare for *all* patients. [32, 46] More specifically, the WHO developed a framework to achieve the desired health outcomes by defining that the quality of maternity care is depending on two inter-linked dimensions, namely the *provision* and the *experience* of care. [32, 46, 47] Indeed, the women's narratives in this intercultural context, confirmed the importance of the two-dimensional attention towards the provision of care along the necessity of effective communication, respectful attitudes and the social and emotional support of caregivers. [32, 46–48]

As such, one could argue that, also within this intercultural setting, the WHO framework and a person-centred approach would be a good starting point to improve the quality in maternity care. Within the person-centered approach, caregivers are encouraged to collaborate with patients to co-design and deliver personalized care, which includes the caregiver's responsiveness towards the patient's preferences, needs and (cultural) values. [49] According to the women's narratives, this 'collaborative' relational care process with caregivers was indeed crucial to their care experiences.

However, our results confirm that providing person-centered care is even more challenging in an intercultural setting since different cultural values and preferences from both patients and caregivers come together *within* the relational process. [50] Starting only from the patient's cultural context might hold the risk of an overemphasis on finding practical solutions for a set of cultural practices, beliefs, values or traditions. In accordance with previous research, we argue that practical solutions to visible religious and cultural tensions can never merely be the answer to the question on how to provide appropriate intercultural care. [4, 13, 17, 51]

Because by doing so, the concept and practice of intercultural care would be predominantly understood as a 'technical art' instead of a 'moral practice'. [4, 51, 52] On the contrary, we have to understand care as the relational process of care-giving and care-receiving in which all involved seek to find a dignified answer to a situation of human vulnerability. [51, 52]

The nature of this relational process in our findings is consistent with Joan Tronto's ethical model, wherein she defined four essential dimensions of the care process: *caring about* (attentiveness towards the patient's needs), *taking care of* (taking responsibility to answer the needs), *care-giving* (the actual work be done) and *care-receiving* (the evaluation on how well the needs were met). [53, 54] In addition to these four dimensions of Tronto, we could argue that the two-dimensional attention of caregivers towards '*Ereignis'* and '*Erlebnis*' is needed in each of the four dimensions of care.

Building on Tronto, Martinsen suggest that in a medical context, it might be useful to distinguish between '*taking care of*' and '*caring for'*. [55] As such, she pointed out that it might be difficult for caregivers to go beyond '*taking care of*' the patients' needs in a technical way. She stressed the need to '*care for*' the patients also by an empathic engagement [55]:

> *"Why is it hard for the physician to simply care for the patient without hiding behind some kind of procedure? Why is it difficult to just sit down and hold the patient's hand, which may be the most appropriate thing to do in this situation?"* (p.114)

Our findings reveal that being aware of this tension might even be more crucial in the maternity care setting in which caregivers have to deal with the extra vulnerability of childbearing women in a post-migration context. [28, 29] Since only few complications occurred, most of the women placed lower emphasis on 'taking care of' in a more technical sense than on 'caring for' since being pregnant and giving birth were in their views a rather natural process that placed them in a special position in which they needed support in an emphatic and attentive way.

This also referred to the field of tension in the Belgian maternity care since care in this context, is predominantly obstetric-led with a high emphasis on the technical aspects and on the pathological potential of being pregnant and giving birth. [56] The women appreciated the high technical competence of caregivers and were grateful for 'the good outcome' because of this safe, high quality care. This gratefulness was in accordance with a previous study on the experiences of Flemish women in Belgian maternity care. [45]

This starting point in the obstetric-led maternity care, however, caused difficulties when the support of caregivers was limited to an *uncaring* or *protocolized* way of 'taking care of' the women. In these cases, a gap was noticed between the women who valued a personal relationship with caregivers and the caregivers who kept a 'professional distance'. Although this professional distance is in accordance with the tradition in medicine of 'staying neutral', in the women's narratives this was not interpreted as a 'neutral' value but rather as a caregivers' unwillingness to '*care for*' them in a personal way. [55] In talking about caregivers who were not emotionally available for them, some women even doubted if this distance was being caused by their Turkish or Moroccan descent, their veil or to (perceived) language problems. In such cases, the professional distance of caregivers was interpreted against the background of this 'ghost of discrimination'.

Our results show that providing intercultural care is being challenged not only because of culturally different views on illness and care or by caregivers who are causing harm (e.g. by discriminating attitudes). On the contrary, most harm took place in *uncaring* care or *protocolized* care when the women were dealing with a *lack* of care (especially towards '*Erlebnis*'). As suggested by Martinsen, our results confirm the importance of interpreting harm as a relational

harm caused by a lack of care. [55] Also here, starting from a person-centred approach can be of merit to avoid 'relational' harm especially in an intercultural setting since it focuses on a 'collaborative' relationship and on caring 'with' patients rather than on 'caring for' or 'taking care of' patients. [49] Remains to be asked how we should understand the role of the women's culture within this relational process of care?

**Culture within the care relationship: A shifting 'cultural' perspective?.**   The women's narratives reveal that culture could not be reduced to a well-defined list of common cultural aspects, nor as being separate from religious, social and psychological dimensions. The women's experiences were also not only embedded in one delineated culture (e.g. either 'Muslim' or 'Flemish' culture) since the women dynamically 'moved between' at least two interwoven cultural contexts depending on their own unique migration process. Moreover, our results show that a distinction between the experiences of women from a Turkish or Moroccan origin did not came to the front in our results. In this regard, we also did not find significant differences between the first, second or third generation of Muslim women, which resonates with previous research in which no differences were found between the health beliefs of first and second generation of Moroccan Muslim women in Belgium. [57]

A such, cultural issues that came to the front cannot be interpreted as being solely belonging to 'a Muslim culture'. In this, our study confirms the suggestion of Kleinman & Benson that it is not feasible nor desirable to manage cultural issues in healthcare by making a list of cultural values, beliefs or practices that has to be taking into account when caring for a specific group of patients. [13, 58] For instance, in talking about the special position of being a childbearing woman, one could easily argue that this could count for all women since being pregnant and giving birth is an important life-changing event for all.

Nevertheless, our results did show that the cultural context of the women is an essential meaning system that is part of the relational care process, which adds another layer of emotional, moral or symbolic meaning to the interactions between the women and their caregivers as well as to the women's expectations and interpretations of care. Similar findings regarding this role of culture as an additional layer to the interpersonal negotiation of care between caregivers and patients has been discussed by Broom et al. [59]

As such, our results show the importance of understanding 'culture' as being inter-relational and dynamic rather than as being an isolated, static, individualized entity. Culture here, can be understood in accordance with the definition of Kleinman & Benson [13]:

'*Culture is not a single variable, but rather comprises multiple variables, affecting all aspects of experience. Culture is inseparable from economic, political, religious, psychological and biological conditions. Culture is a process through which ordinary activities and conditions take on an emotional tone and a moral meaning for participants.* [. . .] *Cultural processes frequently differ within the same ethnic or social group because of differences in age cohort, gender, political association, class, religion, ethnicity, and even personality.*'

As such, our results confirm the findings of recent international research that discusses the necessity of shifting away from an individualized conceptualization of culture in healthcare. [59, 60, 61]

In the cultural competence models as well as in healthcare services, the culture of ethnic minority patients is mostly interpreted as an individual deficit that needs to be taken care of by caregivers. [50, 63, 64] Our results, on the contrary, showed that a shift is needed towards the understanding of culture as being part of the relational care process and thus as emerging *from within* the interaction between caregivers and patients instead of seeing it as an entity that

stands outside these interactions. Here, the findings add more insight into the importance of a shifting perspective on 'culture' from being an 'individual culture-in-isolation' towards a notion of culture as being inter-relational and emerging from within the various care *interactions* with caregivers. [59, 60, 61]

The major practical implication of this notion is the fact that we cannot start from cultural knowledge about a specific isolated culture to handle cultural issues or difficulties in health care practice. On the contrary, providing culturally appropriate care has to start from the establishment of a qualitative and meaningful care relationship *along with* the recognition of the extra emotional and moral meaning that culture adds to the care expectations, interactions and the way in which the care receivers cope with their experiences.

## Strengths of the study

This is the first qualitative study exploring the experiences of Muslim women from Turkish and Moroccan descent who gave birth in a maternity ward in Flanders, Belgium. [45] Empirical evidence on the experiences of ethnic minority women in maternity care worldwide, confirms our findings on the importance of the quality of the care relationship, the influence of communication, the influence of the caregiver's attitudes, the importance of the women's involvement in the decision-making as well as the caregivers' responsiveness towards the women's expectations, (cultural) wishes and needs. [27, 28, 30, 31, 34]

Adding to existing evidence, our study provides new insight into the underlying dynamics in these care relationships by discussing the complex interplay of two essential dimensions of the care process (*Ereignis* and *Erlebnis*) and the two-dimensional attention of caregivers towards these dimensions. Moreover, our results explained the way in which this complex interplay is embedded in the women's cultural context. As such, it adds nuanced insight into the way in which the women's cultural context gives an emotional, moral and symbolic meaning to their care expectations, wishes and needs. Together, the interrelatedness of the two dimensions and the women's cultural meaning system, shaped the manner in which women experienced their care in maternity wards in Belgium. As such, our results provide a framework for critical reflection on the ethical question of providing dignified intercultural care in maternity wards.

## Limitations of the study

As for the limitations of the study, generalizability of the results is limited due to the nature of the study and since our results are grounded in the narratives of the participating Muslim women from Turkish and Moroccan descent who live in the Belgian society as a minority group. Nevertheless, important issues were raised and discussed within our theoretical framework, which can provide a basis for further reflection on maternity care in Belgium and on intercultural care in general. Moreover, the results of this study were confirmed by the results in earlier international research on the perinatal experiences of ethnic minority persons worldwide. As such, we can prudently assume that the concepts that we discussed are also valuable in other settings. Future research is needed in this regard, to evaluate our framework in other settings. As for the maternity care in Belgium, just a few studies exist that explored maternity care experiences. [45, 56, 62–64] Since our study is limited to the perspectives of ethnic minority women, it would be an interesting topic for future research to include the experiences of autochthone women who gave birth in Flemish wards and to include the perspectives of caregivers with experiences in maternity wards in order to compare these angles with the results of our study.

## Possible bias

Furthermore, we were aware of possible bias, especially a possible researcher and selection bias. First, there was the possibility of researcher bias, which needed attention in this study since it concerns a culturally sensitive theme, surrounded with many strong opinions both in contemporary societies and in scientific research. In this regard, we were particularly observant to the fact that the interviewer was not a woman from Turkish or Moroccan descent herself, which included the risk of missing out on important data from an insider point of view, as well as of receiving only socially desirable answers or normative answers and opinions (i.e. on how it *should* be), without insight into the *real* experiences of the women (i.e. how it actually happened). To avoid this, the research group decided to start each interview with a personal introduction of the interviewer by which a start was possible on the common ground of being a woman, a wife, a mother and raised in Limburg. Social desirability and answers from a normative stance were avoided by this, since the women could talk freely by taking this start 'from one mother to another'. Analogously, we applied built-in guarantees, such as reflexivity and bracketing to ensure the trustworthiness of the data. An ongoing reflective journal was kept by the interviewer (LD) using thick descriptions of the interviews and of all the choices made during recruitment, data collection and analysis. [35] The journal also included meanings given to the data and notes on how the interviewer was thinking about the subject before, during and after the study (bracketing). To reduce the risk of missing out on data because of interpreter bias, we cross-checked the translations with an independent interpreter and marked the data in which possible opinions or influence of the interpreter crept in.

Secondly, we were aware of possible selection bias, since all the women had a delivery with positive outcome (e.g. no stillborn babies). As such, we have no results on the overall care experiences of Muslim women with a tragic outcome. Nevertheless, we did perform theoretical sampling, which resulted in a relevant variety of women's' characteristics and hospital characteristics. As such, we reached a rich dataset of in-depth experiences, which allowed us to describe the intercultural care experiences of Muslim women in the context of maternity wards in Belgium in a highly nuanced manner. We reached data saturation on the concepts described in this paper.

Other strategies to ensure the trustworthiness of the findings were: (1) *Researcher triangulation* (analysis performed by two researchers LD and YD). (2) *Data triangulation* by space triangulation (women giving birth in six different hospitals) and person triangulation (the women differ in origin, age of migration, number of children, complications during their hospital stay, language ability, etc.). (3) *Peer review*: frequent meetings with the multidisciplinary research (YD, LD, BDC, and CG) team to critically compare and modify the results. (4) *Peer debriefings* with external experts at several stages in the research project.

## Conclusion

The findings reveal that the quality of intercultural care predominantly depends on the nature and quality of *care relationships* between ethnic minority patients and caregivers rather than on the way in which cultural questions and tensions are being handled or dealt with. As such, the importance of establishing a *meaningful* care relationship should be priority in providing intercultural care. Therefore, a shift in perspective on culture from being an 'individual culture-in-isolation' towards the concept of culture as being 'inter-relational' and dynamically emerging *from within* these care relationships is needed.

## Supporting information

**S1 Appendix. Examples of interview questions.**
(DOCX)

**S2 Appendix. Ethical considerations interview process.**
(DOCX)

## Acknowledgments

We are grateful to all the Muslim women who participated in the study. Furthermore, we thank all the key persons for their support in the process of data collection and the interpreters for their translation. We also thank Prof. Dr. Patrick Meurs for his key role in the data analysis and Dr. Chaïma Ahaddour and Dr. Delphine Jacobs for their supportive assistance during several stages of the study.

## Author Contributions

**Conceptualization:** Liesbet Degrie, Chris Gastmans, Yvonne Denier.

**Data curation:** Liesbet Degrie, Yvonne Denier.

**Formal analysis:** Liesbet Degrie, Bernadette Dierckx de Casterlé, Chris Gastmans, Yvonne Denier.

**Funding acquisition:** Chris Gastmans, Yvonne Denier.

**Investigation:** Liesbet Degrie, Chris Gastmans, Yvonne Denier.

**Methodology:** Liesbet Degrie, Bernadette Dierckx de Casterlé, Chris Gastmans, Yvonne Denier.

**Project administration:** Yvonne Denier.

**Supervision:** Bernadette Dierckx de Casterlé, Chris Gastmans, Yvonne Denier.

**Validation:** Bernadette Dierckx de Casterlé, Chris Gastmans, Yvonne Denier.

**Writing – original draft:** Liesbet Degrie, Yvonne Denier.

**Writing – review & editing:** Bernadette Dierckx de Casterlé, Chris Gastmans, Yvonne Denier.

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
