## [Decision Letter · Decision Letter 0]

17 Oct 2019

PONE-D-19-20242

‘Can you please hold my hand too, not only my breast?’ The experiences of Muslim patients from Turkish or Moroccan descent giving birth in maternity wards in Belgium.

PLOS ONE

Dear Dr. Denier,

Thank you for submitting your manuscript to PLOS ONE. After careful consideration, we feel that it has merit but does not fully meet PLOS ONE’s publication criteria as it currently stands. Therefore, we invite you to submit a revised version of the manuscript that addresses the points raised during the review process.

We would appreciate receiving your revised manuscript by Dec 01 2019 11:59PM. To enhance the reproducibility of your results, we recommend that if applicable you deposit your laboratory protocols in protocols.io, where a protocol can be assigned its own identifier (DOI) such that it can be cited independently in the future. For instructions see: http://journals.plos.org/plosone/s/submission-guidelines#loc-laboratory-protocols

We look forward to receiving your revised manuscript.

Kind regards,

Virginia E. M. Zweigenthal

Academic Editor

PLOS ONE

Journal Requirements:

1. We note that you have indicated that data from this study are available upon request. PLOS only allows data to be available upon request if there are legal or ethical restrictions on sharing data publicly. For information on unacceptable data access restrictions, please see http://journals.plos.org/plosone/s/data-availability#loc-unacceptable-data-access-restrictions.

Additional Editor Comments (if provided):

Many thanks for an excellent article on of patient experiences of childbirth care. Your nuanced analysis contributes to the literature on this subject, particularly around care for women from culturally different communities. Your thorough description of the methods would be a good reading for students learning qualitative methods.

There are a few minor suggested changes, from reviewer 1. I also found on line 736 a typo 'Death' should be 'dead'.

Many thanks,

Dr Virginia Zweigenthal

Reviewers' comments:

Reviewer's Responses to Questions

**Comments to the Author**

1. Is the manuscript technically sound, and do the data support the conclusions?

Reviewer #1: Yes

Reviewer #2: Yes

2. Has the statistical analysis been performed appropriately and rigorously? 

Reviewer #1: N/A

Reviewer #2: N/A

3. Have the authors made all data underlying the findings in their manuscript fully available?

Reviewer #1: Yes

Reviewer #2: Yes

4. Is the manuscript presented in an intelligible fashion and written in standard English?

Reviewer #1: Yes

Reviewer #2: Yes

5. Review Comments to the Author

Reviewer #1: Thank you very much for asking me to review this interesting, detailed and well written qualitative research paper the experiences of Muslim patients from Turkish or Moroccan descent giving birth in maternity wards in Belgium. It is an important topic for maternity care in the Belgium and beyond, and the authors have provided a thoughtful and nuanced analysis which provide a great insight to addresses the issue of intercultural care.

I have a few comments which would help the reader to interpret the paper, don’t worry them may seem a lot but they are just small changes:

1. The title reads ‘Muslim patients from Turkish or Moroccan descent…’ I think you need to change to ‘and’ and not ‘or’ because you involved both in the study.

2. Also I am not sure if the term ‘Patients’ is the correct term to use, these women are giving birth in a hospital setting but they are not necessary ill patients, childbirth is not an illness. So I suggest you call them ‘mothers’, or ‘women’ utilizing hospital settings for childbirth.

3. Method section – line 143. You mention that study included women that were ‘hospitalized in maternity wards’. This needs more clarity- are we talking about women who had clinical complication and needed to be clinically hospitalized or are we talk about women that use a hospital setting as it is the normal setting for women to utilize during childbirth? Be mindful of the terms you use. Also the inclusion criteria- need more clarity- was it open to all women, such to those that had virginal birth, complications, caesarean section delivered, instrumental birth, or other complex health needs.

4. Method section – line 156. You mention purposive sampling to recruit first participants: how many? Line 157 – you mention after analysis these interviews; again how many before you made the discussion change approach.

5. Method section – line 159 ‘after a few interviews’ how many?

6. Method section – line 174 to 191 – most of the information in this section repeats what is already reflected in table 2. So I suggest that you have the characteristic of participants in the table and you don’t need to have that paragraph, just add the details to the table.

7. Method section – line 242 table 3. I suggest move to appendixes, you have provided enough details about the ethical process, so no need for the table in text.

8. Result section- line 279 you mention ‘Muslim women’ and you then you change terms throughout the section using the terms ‘participants’ , ‘patients’ I suggest you need to be constant in the term you use. If you are going to call them women for example then keep it as women throughout.

9. Result section- line 292 you used the term ‘vulnerable bodies’, I suggest you change this term. Caregiver are there to provide care for the women and not only their body.

10. Result section – line 337 to 339. Sentence needs to be toned down ‘obey’ can be ‘follow’ also women ‘felt’ that there was ‘no regards for their own voice’ remember this is how women felt, you are not stating facts but you reporting the women’s narratives.

11. Result section – line 374. ‘recalled events in which …’ I suggest you include ‘women felt that various caregivers treated then in…’

12. Discussion section – it’s best not to start the discussion with the strengths and limitations and possible bias. I suggest you move these sections to the end of the discussion before the conclusion. Start discussion from line 813.

Looking forward to seeing the revised version.

Best of luck

Reviewer #2: This is an excellent study of the experiences of ethnic minority women accessing maternity care in Belgium. My only (very minor) quibble is that the labels for the four types of care aren't always distinct or precise for every use ('ambiguous', for example, doesn't seem like the most precise way to capture the fourth domain, and there were elements of ambiguity in the other domains). But aside from this minor concern, the methodology was clear and strong and the analysis rich.

6. PLOS authors have the option to publish the peer review history of their article (what does this mean?). If published, this will include your full peer review and any attached files.

Reviewer #1: No

Reviewer #2: No

---

## [Author Response · Author response to Decision Letter 0]

24 Apr 2020

Rebuttal letter - Response to Reviewers

General reply from the authors: 

We sincerely thank the editor and both reviewers for their supportive and useful comments. We are grateful for the positive feedback and suggestions and read them as a confirmation of the value of our research project to the literature on the subject of care for women from culturally different communities. We also very much appreciate the editor’s comment that our methods section will be fruitful for students learning qualitative methods. Thank you.

Following the questions and suggestions of the editor and reviewers, we have accommodated the paper to all made concerns. In the point-by-point response below, you can find our specific answers and references to the changes made in the manuscript. A marked-up copy, which highlights the changes made to the original version is also available as ‘Revised Manuscript with Track Changes’.

First, we start with the general comment regarding Journal Requirements.

Journal Requirements: 

Requirement #1: 

• http://www.journals.plos.org/plosone/s/file?id=wjVg/PLOSOne_formatting_sample_main_body.pdf and

• http://www.journals.plos.org/plosone/s/file?id=ba62/PLOSOne_formatting_sample_title_authors_affiliations.pdf

Reply from the authors: 

Thank you. We carefully checked the requirements and templates and made the necessary adjustments in the revised manuscript. Hopefully we have fulfilled all the style requirements. Please let us know, when something would still be missing.

Requirement #2: 

We note that you have indicated that data from this study are available upon request. PLOS only allows data to be available upon request if there are legal or ethical restrictions on sharing data publicly. For information on unacceptable data access restrictions, please see http://journals.plos.org/plosone/s/data-availability#loc-unacceptable-data-access-restrictions.

Reply from the authors: 

Thank you for pointing our attention to further clarification of our data availability statement, which we are happy to provide. Data availability is indeed restricted for ethical and legal reasons (§a of the above-mentioned prompts). In the revised manuscript, we have added the following explanation in order to provide more detailed information on this restriction (lines 1045-1057): 

“Availability of data and material 

The dataset of the study consists of 24 in-depth and detailed interviews (transcribed ad verbatim) with women from Turkish or Moroccan descent, inquiring after their experiences during childbirth in the hospital context. The interviews contain highly confidential, sensitive and potentially identifying patient information. These interviews were only possible on the condition of strict confidentiality, as agreed upon in the informed consent procedure and signed IC forms. Therefore, we are not able to make the dataset publicly available via open access, because of ethical and legal restrictions. This was also accorded by the Social and Societal Ethics Committee (SMEC) of KU Leuven, which granted approval of the research project (G 2016 03 531). In order to guarantee trustworthiness and reproducibility of the results, we applied multiple strategies as explained in the Methods and Discussion section. Requests for data information and access can be sent at all times to the Social and Societal Ethics Committee (SMEC) of KU Leuven (e-mail: smec@kuleuven.be research project: G 2016 03 531). See also: https://www.kuleuven.be/english/research/ethics/committees/smec#section-0”

We hope this provides sufficient clarification of the reasons why the data availability of this project is restricted. 

Comments from the editor: 

Many thanks for an excellent article on patient experiences of childbirth care. Your nuanced analysis contributes to the literature on this subject, particularly around care for women from culturally different communities. Your thorough description of the methods would be a good reading for students learning qualitative methods.

There are a few minor suggested changes, from reviewer 1. I also found on line 736 a typo 'Death' should be 'dead'.

Reply from the authors:

Thank you very much for your general appreciation of our paper. We have corrected the typo. 

Comments from the reviewers 

Reviewer #1: 

Thank you very much for asking me to review this interesting, detailed and well written qualitative research paper on the experiences of Muslim patients from Turkish or Moroccan descent giving birth in maternity wards in Belgium. It is an important topic for maternity care in the Belgium and beyond, and the authors have provided a thoughtful and nuanced analysis which provide a great insight to addresses the issue of intercultural care.

I have a few comments which would help the reader to interpret the paper, don’t worry them may seem a lot but they are just small changes:

Reply from the authors:

Thank you very much for your warm compliments and apt suggestions for improving the paper. Below, you can find our detailed reply. In the manuscript (‘Revised Manuscript with Track Changes’) you can see all the changes in highlight. 

Reviewer #1:

The title reads ‘Muslim patients from Turkish or Moroccan descent…’ I think you need to change to ‘and’ and not ‘or’ because you involved both in the study.

Reply from the authors:

Thank you for your apt comment. We went through the paper in full and made the suggested changes.

Reviewer #1:

Also I am not sure if the term ‘Patients’ is the correct term to use, these women are giving birth in a hospital setting but they are not necessary ill patients, childbirth is not an illness. So I suggest you call them ‘mothers’, or ‘women’ utilizing hospital settings for childbirth.

Reply from the authors:. 

That’s correct. We agree with your suggestion, and changed the term ‘patient’ or ‘patients’ into ‘woman’ or ‘women’ when the women’s narratives were involved. 

We kept the term ‘patient’ or ‘patients’ when we refer to the literature in general, most particularly when we refer to studies/research projects/papers/data that refer to ethnic minority ‘patients’ themselves.

Reviewer #1:

Method section – line 143. You mention that study included women that were ‘hospitalized in maternity wards’. This needs more clarity- are we talking about women who had clinical complication and needed to be clinically hospitalized or are we talk about women that use a hospital setting as it is the normal setting for women to utilize during childbirth? Be mindful of the terms you use. Also the inclusion criteria- need more clarity- was it open to all women, such to those that had virginal birth, complications, caesarean section delivered, instrumental birth, or other complex health needs.

Reply from the authors:

Thank you for your apt remark. Accordingly, we made the following changes in the manuscript (lines 142-146) in order to makes sure that the inclusion criteria are more clear: 

“Participants were included when they considered themselves as Muslim women from Turkish and Moroccan descent and were hospitalized in maternity wards (considered as a normal setting for women to utilize during childbirth) in the province of Limburg during the last 3 years. As for the type of birth, selection was open to varying types (natural births, first and following births, caesarean, instrumental, and more complex types with specific health needs). 

Reviewer #1:

Method section – line 156. You mention purposive sampling to recruit first participants: how many? Line 157 – you mention after analysis of these interviews; again how many before you made the discussion change approach.

Reply from the authors:

Our purposive sampling consisted of 6 to 8 interviews. Regarding your second question, this concerned the same amount of interviews. We have added this clarification to the revised manuscript (lines 158-159). 

Reviewer #1:

Method section – line 159 ‘after a few interviews’ how many?

Reply from the authors:

This was after 6 interviews. We have added this information to the revised manuscript (line 161).

Reviewer #1:

Method section – line 174 to 191 – most of the information in this section repeats what is already reflected in table 2. So I suggest that you have the characteristic of participants in the table and you don’t need to have that paragraph, just add the details to the table.

Reply from the authors:

Thank you. We have removed redundant information from the paragraph. 

Reviewer #1:

Method section – line 242 table 3. I suggest move to appendices, you have provided enough details about the ethical process, so no need for the table in text.

Reply from the authors:

Thank you. We moved it to the appendices (Supporting Information, ‘S2 Ethical Considerations Interview Process’). 

Reviewer #1:

Result section- line 279 you mention ‘Muslim women’ and you then you change terms throughout the section using the terms ‘participants’ , ‘patients’ I suggest you need to be constant in the term you use. If you are going to call them women for example then keep it as women throughout.

Reply from the authors:

Thank you. We changed the terms throughout the text, using the term ‘women’, when it concerned the participants of our study. When we refer to ‘recruitment of participants’ in general, as a general term in empirical research, we kept the term ‘participants’. 

Reviewer #1:

Result section- line 292 you used the term ‘vulnerable bodies’, I suggest you change this term. Caregiver are there to provide care for the women and not only their body.

Reply from the authors:

Correct. We changed it. 

Reviewer #1:

Result section – line 337 to 339. Sentence needs to be toned down ‘obey’ can be ‘follow’ also women ‘felt’ that there was ‘no regards for their own voice’ remember this is how women felt, you are not stating facts but you reporting the women’s narratives.

Reply from the authors:

Thank you. This is an important change, that we are happy to make, since it indeed refers to the way in which the women felt, what we learn from their narratives (and not as a fact). Thank you! We made the necessary changes in order to make this more clear (cf. lines 340-344).

Reviewer #1:

Result section – line 374. ‘recalled events in which …’ I suggest you include ‘women felt that various caregivers treated then in…’

Reply from the authors:

Thank you. This is a similar important change in nuance, that we were happy to make (cfr. lines 376-378). 

Reviewer #1:

Discussion section – it’s best not to start the discussion with the strengths and limitations and possible bias. I suggest you move these sections to the end of the discussion before the conclusion. Start discussion from line 813.

Reply from the authors:

Thank you very much. We also made this change in the revised version. 

Reviewer #2: This is an excellent study of the experiences of ethnic minority women accessing maternity care in Belgium. My only (very minor) quibble is that the labels for the four types of care aren't always distinct or precise for every use ('ambiguous', for example, doesn't seem like the most precise way to capture the fourth domain, and there were elements of ambiguity in the other domains). But aside from this minor concern, the methodology was clear and strong and the analysis rich.

Reply from the authors:

Thank you very much for your appreciation of our study. Indeed you are right in stating that the labels are less precise than might have been hoped for. 

The labelling is the result of a long and interdisciplinary process (2-3 years), with a lot of internal discussions within the research team, as well as with the consulted external experts during various peer reviewing and peer debriefing meetings. As such, we trust that they adhere to the complexity and highly nuanced nature of the intercultural care process in all its variety and ambiguity. Most likely, it is also because of the fact that this care process is so complex, that the various labels cannot point at a clear-cut clarification of the domains. Having been able, however, to thoroughly discuss the labels with internal and external researchers makes us confident of their added value to the literature and practice of intercultural hospital care. 

General reply from the authors: We have changed our paper in line with the comments received from the editor and the two reviewers. We are grateful and pleased with the improved result. Thank you!

---

## [Decision Letter · Decision Letter 1]

29 Jun 2020

‘Can you please hold my hand too, not only my breast?’ The experiences of Muslim women from Turkish and Moroccan descent giving birth in maternity wards in Belgium.

PONE-D-19-20242R1

Dear Dr. Denier,

We’re pleased to inform you that your manuscript has been judged scientifically suitable for publication and will be formally accepted for publication once it meets all outstanding technical requirements.

Kind regards,

Virginia E. M. Zweigenthal

Academic Editor

PLOS ONE

Additional Editor Comments (optional):

Dear Colleagues,

many thanks for your revision which do address the issues raised in the reviews.

The article addresses difficult issues of intercultural care in maternity care, giving insight on dimensions of care that need attention. It is well written, thorough and rigorous. Consequently, we believe that the article merits publication.

Best wishes,

Dr Virginia Zweigenthal

Reviewers' comments:

Reviewer's Responses to Questions

**Comments to the Author**

1. If the authors have adequately addressed your comments raised in a previous round of review and you feel that this manuscript is now acceptable for publication, you may indicate that here to bypass the “Comments to the Author” section, enter your conflict of interest statement in the “Confidential to Editor” section, and submit your "Accept" recommendation.

Reviewer #2: All comments have been addressed

2. Is the manuscript technically sound, and do the data support the conclusions?

Reviewer #2: Yes

3. Has the statistical analysis been performed appropriately and rigorously? 

Reviewer #2: N/A

4. Have the authors made all data underlying the findings in their manuscript fully available?

Reviewer #2: Yes

5. Is the manuscript presented in an intelligible fashion and written in standard English?

Reviewer #2: Yes

6. Review Comments to the Author

Reviewer #2: (No Response)

7. PLOS authors have the option to publish the peer review history of their article (what does this mean?). If published, this will include your full peer review and any attached files.

Reviewer #2: No

---

## [Editor Report · Acceptance letter]

8 Jul 2020

PONE-D-19-20242R1 

‘Can you please hold my hand too, not only my breast?’ The experiences of Muslim women from Turkish and Moroccan descent giving birth in maternity wards in Belgium. 

Dear Dr. Denier:

I'm pleased to inform you that your manuscript has been deemed suitable for publication in PLOS ONE. Congratulations! Your manuscript is now with our production department. 

Kind regards, 

on behalf of

Dr. Virginia E. M. Zweigenthal 

Academic Editor

PLOS ONE